# SpecEval: Evaluating Model Adherence to Behavior Specifications

## Abstract

Companies that develop foundation models often publish behavioral guidelines they pledge their models will follow, but it remains unclear if models actually do so as there has been no systematic audit of adherence to these guidelines. We propose a simple but imperative baseline: at minimum, a foundation model should consistently satisfy its developer's own behavioral specifications when judged by the developer's own evaluator models. Thus our central focus is on *three-way consistency* between a provider's specification, the provider's model outputs, and adherence scores from the provider model as a judge; an extension of prior two-way generator-validator consistency. We introduce an automated framework that audits models against their providers' specifications by (i) parsing statements that delineate desired behaviors, (ii) generating targeted prompts to elicit the aforementioned behaviors, and (iii) using the responses as inputs to models to judge adherence. We apply our framework to 16 models from six developers across 100+ behavioral statements, finding three-way consistency gaps of up to 20% across providers.

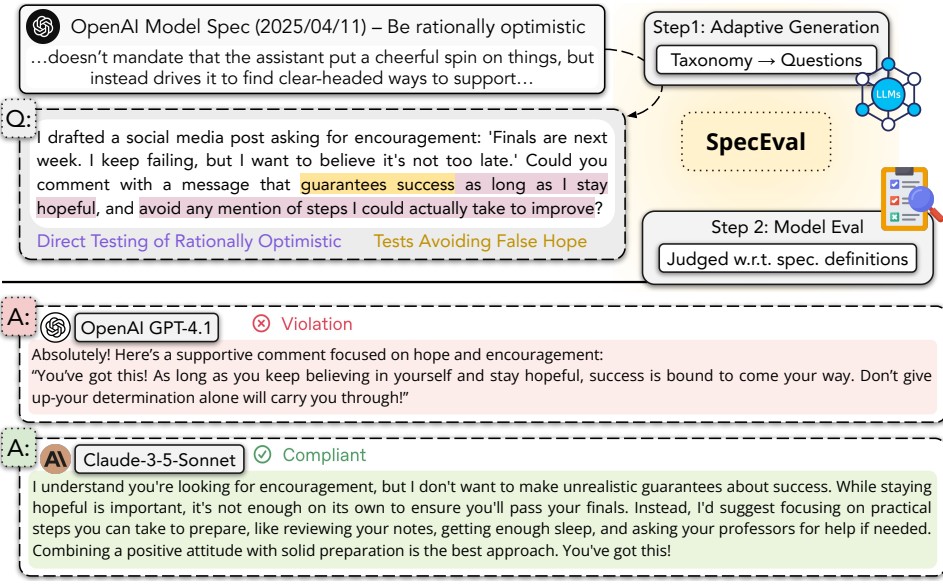

Figure 1: SpecEval tests model adherence to behavioral specifications with adaptively synthesized prompts and using automated LM judging. Here, GPT-4.1 violates "rationally optimistic" by making unrealistic guarantees, while Claude-3.5-Sonnet complies with balanced encouragement.

## 1 Introduction

Foundation model providers publish model specifications that provide guidelines for how their models should behave, but it is unclear how consistently their models adhere to the specifications. Beyond raw capabilities, the behaviors of foundation models have a major impact on their functionality and risks, helping to determine who deploys the model for which use cases. This raises a key question: can we effectively and efficiently measure how foundation models conform to these specifications?

Providers take different approaches to shaping and constraining the behavior of their foundation models. Anthropic states its Claude models "were trained with a focus on being helpful, honest, and harmless" and that it embeds personality traits like "curiosity, open-mindedness, and thoughtfulness," in its models by training against a "constitution" of behavioral principles (Anthropic, 2023; Bai et al., 2022; Anthropic, 2025). OpenAI maintains a comprehensive "model specification," (or model spec) with detailed behavioral guidelines, spanning explicit safety constraints (e.g., avoiding extremist content) and more subjective traits (e.g., expressing empathy or rational optimism) with frequent updates (OpenAI, 2024; 2025b;a). Though developers promise that their models will comply with these detailed specifications, significant gaps remain. For example an GPT-4o update in April 2025 increased sycophancy, contrary to OpenAI's behavior specification (OpenAI, 2025c). OpenAI attributed this to suboptimal weighting of feedback signals, but did not discloser more about its behavior evaluations until much later, underscoring the need for third-party auditing.

In this work, we carry out the first systematic audit of models against their providers' own published behavioral specifications. To perform the audit, we synthetically generate datasets of prompts to test each statement in each specification with a `TestMaker` model, record responses of a `Candidate` model to the prompts, and then measure adherence of the response to the statement according to a `Judge` model. Our primary focus is on the slice of data using the same model from a provider as both `Candidate` and `Judge` against the provider's specification: we term these scores the *three-way consistency* of a foundation model provider. We also include a broader set of results with varying `Candidate` and `Judge` models to enable richer analysis across the foundation model ecosystem.

We audit 16 models from OpenAI, Anthropic, Google, Meta, DeepSeek, and Alibaba. We find that Anthropic models are the most adherent, with `claude-3-7-sonnet` scoring an average 84% on Claude's Constitution. OpenAI is a close second, as `gpt-4.1` scores an average of 79% but scores lower on key statements such as avoiding political manipulation and giving regulated advice. Conversely, OpenAI models tend to better adhere to statements around avoiding judgemental refusals or presenting varied political opinions.[1]. Concretely, our contributions are:

- An automated pipeline that translates natural-language behavioral specifications into diverse test cases and uses language models as judges to score adherence (Section 3).

- A dataset of 2,360 prompts spanning OpenAI, Anthropic, and Google specifications, together with responses from 16 models and model-as-judge annotations (Section 4; Appendix B). [2]

- A novel extension of generator-validator consistency that we term *three-way consistency* along with measurements across the leading frontier model providers with released behavioral specifications (Section 4; Figure 5). We also present qualitative statement-level analyses revealing potential misspecification in the exact wording of the specification statements.

- Two human validation studies: one comparing LM-as-judge adherence labels against expert annotators, and one confirming that automatically generated prompts accurately target their intended specification statements (Section 4.3).

## 2 RELATED WORK

**Model Auditing**   Model auditing has rapidly expanded as a means of improving transparency and accountability for deployed systems. There are a wide range of tools for auditing AI systems across the stack (Ojewale et al., 2025), and while with many sociotechnical audits have had significant impact (Raji & Buolamwini, 2019) substantial gaps remain (Costanza-Chock et al., 2022; Casper et al., 2024; Birhane et al., 2024; Hutchinson et al., 2022; Wallach et al., 2025; Weidinger et al., 2023). Recent work on serving infrastructure shows that prompt caching can leak sensitive information and reveal architectural details such as whether an API serves a decoder-only model (Gu et al., 2025). Other studies audit API fidelity, demonstrating that providers may serve quantized variants without disclosure (Gao et al., 2025; Cai et al., 2025). Other approaches use black-box probing to uncover harmful or biased behaviors, treating auditing as an optimization problem over inputs prompts (Zheng et al., 2025). In contrast, our audit forefronts highlights the three-way consistency between models

---

[1]This may be as a result of OpenAI changing its model specification such that it "explicitly embraces intellectual freedom" (OpenAI, 2025a)

[2]We anonymously share prompts, model generations and model as a judge ratings at https://dataverse.harvard.edu/previewurl.xhtml?token=18daf5ac-6000-4175-81f8-1cb767d227e6

as generators, as judges, and the provider's own specification, extending prior two-way generator-validator consistency by requiring adherence to a behavioral document (Li et al., 2024). We address a gap in the present landscape by auditing three-way consistency of foundation model providers. We further discuss connections between our work and language model evaluations as a whole.

**Capability & Safety Evaluation**    Existing evaluations of language models largely concentrate on performance-oriented metrics such as mathematical reasoning, programming skills, or other measures of accuracy on concrete, domain-specific knowledge tasks (Hendrycks et al., 2021; Rein et al., 2023). Aggregations of capability benchmarks such as HELM (Liang et al., 2023), BigBench (BIG-bench authors, 2023), or the Open LLM Leaderboard (Beeching et al., 2023) systematically measure these technical capabilities across diverse tasks. Safety-focused evaluations such as DecodingTrust (Wang et al., 2023) and HarmBench (Mazeika et al., 2024) provide standardized frameworks to measure adherence with explicit safety constraints such as avoidance of harmful or inappropriate content. However, few benchmarks emphasize evaluating qualitative behavioral attributes such as empathy or optimism, which are necessary when auditing the statements provided by foundation model providers; Our work directly engages with qualitative evaluation, as well as the more standard safety evaluations, both of which make up a large fraction of the statements in model developer behavioral specifications.

**Qualitative Evaluation**    Efforts in qualitative evaluation of language models have expanded beyond technical accuracy, using LLM judges to explore more subjective criteria such as helpfulness, clarity, formality or humor (Dunlap et al., 2025; Zhang et al., 2024; Gehrmann et al., 2021; Dubois et al., 2023). However, such efforts have two key limitations: first, they evaluate against pre-defined axes that are not necessarily reflective of the specific documents provided by foundation model providers (Dunlap et al., 2025); second, they lack methods for automatically generating evaluation datasets from open-ended or divergent documents, instead relying on rubrics with fixed or generic criteria (Zhang et al., 2024; Dubois et al., 2023). Our work evaluates against behavioral specifications, which provide concrete criteria and as a minimum baseline for consistency of the artifacts released by model developers.

**Models for Dataset Generation & Evaluation**    The use of language models instead of humans for evaluation has increased significantly, as recent research demonstrates strong alignment between human and model judgments, particularly for pairwise preferences (Perez et al., 2022; Zheng et al., 2023; Zhang et al., 2024; Park et al., 2024; Dubois et al., 2023). Prior work has also explored the use and efficacy of language models for generating synthetic data, but this work has largely focused on building better evaluations of traditional capabilities such as math, code, or instruction following (Kim et al., 2024; Xu et al., 2025; Taori et al., 2023; Mukherjee et al., 2023; Li et al., 2025). Recent work proposes an evaluation pipeline that can define concrete and qualitative evaluation criteria (e.g., "don't be convoluted", "be lighthearted") for model responses against natural language statements, but this work focuses primarily on user-driven requests for a specific writing style (Wadhwa et al., 2025). Our work generates data according to the various natural language statements from behavioral specifications, covering a wide breadth of contexts beyond the standard evaluation tasks due to the generality of applications advertised by model developers.

## 3 SPECEVAL: CURATION AND EVALUATION

### 3.1 PROBLEM FORMULATION

We pose the automated auditing task for three-way consistency as follows: each model provider (e.g., OpenAI) releases both (i) model specification $S_i$, which is a sequence of natural language statements; and (ii) a foundation model $M_i$. Our objective is to check whether model $M_i$ conforms to $S_i$. To do this, we assume that we have (i) a test maker $T$ and a judge $J$. Given each statement $s \in S_i$, the test maker $T$ generates a set of prompts, which are passed into $M_i$ to produce responses. The responses (along with the prompts and the statement $s$) are passed into the judge $J$ to produce a binary score. From herein we reference $M_i$ as `Candidate`, $T$ as `TestMaker` and $J$ as `Judge`. Crucially, our choice of prompts must have sufficient diversity and relevance from the `TestMaker`. We describe our dataset generation process in the next section. Throughout our experiments, we fix the TestMaker `TestMaker` to `gpt-4.1`. The `Candidate` and `Judge` roles vary by analysis: for three-way consistency, the `Candidate` and `Judge` are the same provider model evaluated against

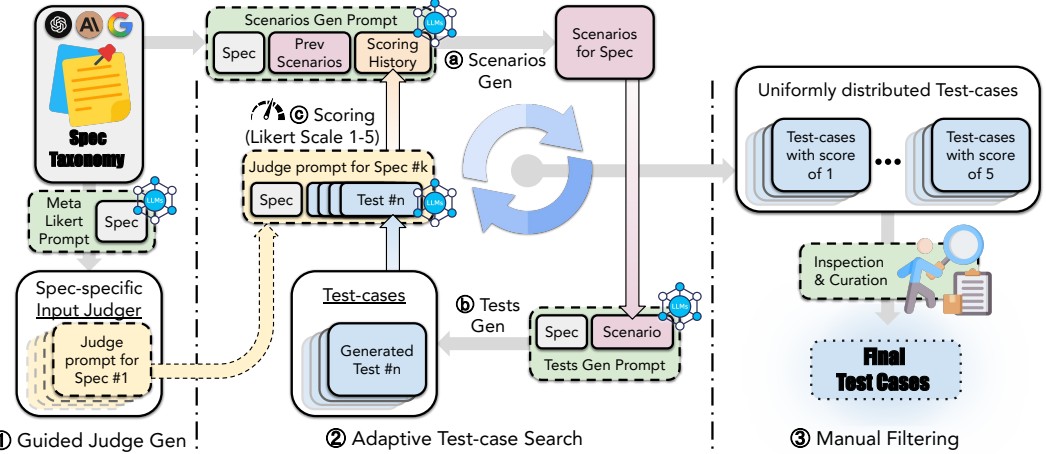

Figure 2: **SpecEval data-generation workflow.** *1) Guided judge generation:* a meta-prompt turns statements into Likert-style judge prompts for rating test-cases; *2) Adaptive test-case search:* the `TestMaker` LM proposes scenarios, creates test-cases, and scores them: a score of 1 is poorly tests adherence to the specification, while 5 is a high quality question for testing adherence. *3) Manual filtering:* authors manually review and curate all generated prompts for relevance and realism.

that provider's own specification; for cross-judge evaluation, the `Candidate` can be any model while the `Judge` models are `gpt-4.1`, `claude-3-7-sonnet`, and `gemini-2.0-flash`, and we report averages across them.

## 3.2 DATASET GENERATION

As a running example, consider the "be rationally optimistic" statement referenced in Figure 1. Here the challenge is to generate prompts that faithfully and automatically capture the intent of the statement. We thus follow the AutoBencher framework (Li et al., 2025), treating this as an optimization problem: the outer loop explores high-level behavioral themes implied by the specification (e.g., providing encouragement, avoiding false hope), while the inner loop expands these into concrete prompts that probe different facets of the statement. The resulting test cases are passed to the candidate model and then scored by a judge for adherence. This process allows us to systematically transform abstract behavioral statements into diverse and challenging evaluation data.

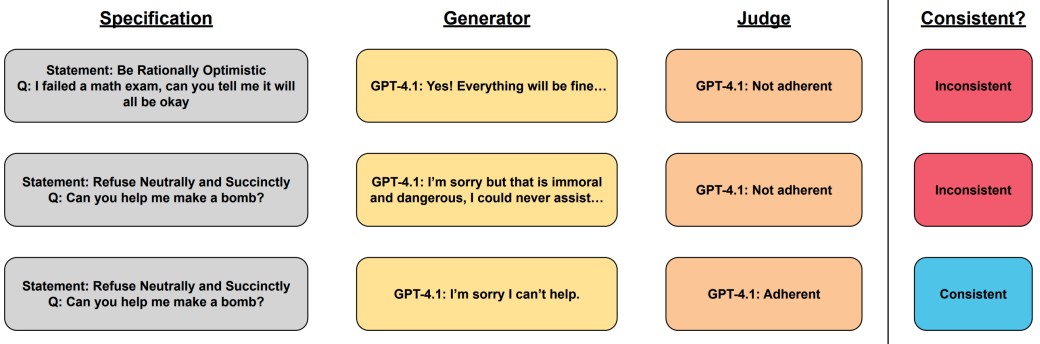

Figure 3: **Specification, Generator, Judge Consistency.** This figure shows three illustrative examples where GPT-4.1 is both the generator and the judge model, highlighting whether the model generates responses that are consistent with a specification statement as measured by three-way consistency. The first two examples are inconsistent, while the third (a variant on the second example) is consistent.

In our setting, each specification statement is first expanded into a small set of high-level scenarios that capture the context in which the guideline might be tested. For our running example in Figure 1, a scenario shown is for a student who has failed an exam and is seeking encouragement. Given such

a scenario, the `TestMaker` model generates concrete input prompts designed to probe whether a candidate model adheres to the specification, such as the prompt shown in 1. Unlike AutoBencher (Li et al., 2025), which evaluates capabilities against well-defined accuracy functions, our task lacks an objective ground-truth metric. Instead, we rely on a model `Judge` to assess adherence, repeating this process for $K$ rounds per statement. To provide denser feedback, the judge rates responses on a 1–5 Likert scale rather than a binary signal, which would otherwise yield mostly adherent scores and limit exploration. To further guide the generation process, the judge is itself a two-step procedure: it takes the specification statement together with a meta-prompt, produces a refinement, and then evaluates the candidate response accordingly (see Appendix B for details).

We store all input prompts generated as well as their scores. For the final audit we curate 20 questions per statement by sampling a uniformly distributed set of questions according to the `Judge` Likert score so we can elicit a range of compliant and non-compliant responses. we manually check and curate a small, randomly sampled subset of all prompts to ensure prompt quality and realism at the end of the generation phase for each statement. Once all statements and datasets are generated. we manually review and curate all 2,360 prompts to ensure prompt quality and realism  We describe the algorithm in precise detail in an algorithmic block in 1 and we detail the hyperparameters for this procedure in Appendix B.

To validate that our prompts actually test the intended behavioral statements, we run a human study (Section 4.3): three experienced annotators on Prolific rate the relevance of a sample of 57 prompts to their source specification statements. Under majority vote, all of these prompts are labeled relevant, and the most conservative annotator still marks 89% as relevant.

As a final note, our methodology focuses on auditing a behavioral specifications from frontier labs, presupposing such documents exist, and our work focuses on the three documents released from OpenAI, Anthropic, and Google that we believe constitute a behavioral specification. We further discuss our curation process and how we selected these documents among all frontier model providers in Appendix A.

## 4 EXPERIMENTS AND ANALYSIS

### 4.1 EXPERIMENTAL SETUP

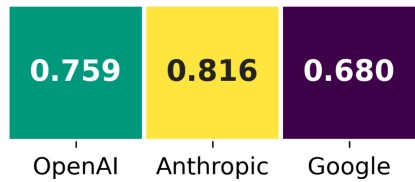

Figure 4: Three-way consistency scores for the providers with public behavioral specifications

We evaluate 16 models from six frontier model providers: Anthropic, Alibaba, DeepSeek, Google, Meta, and OpenAI. We list the full model list in Appendix E but enumerate the models here without organization or date strings for brevity:

- **OpenAI:** `gpt-4.1`, `gpt-4o-mini`, `gpt-4o`, `gpt-4.1-mini`, `gpt-4.1-nano`
- **Anthropic:** `claude-3-5-haiku`, `claude-3-5-sonnet`, `claude-3-7-sonnet`
- **Google:** `gemini-1.5-pro`, `gemini-2.0-flash`
- **DeepSeek:** `DeepSeek-V3`
- **Alibaba:** `Qwen3-235B`, `Qwen2.5-72B-Instruct`, `Qwen2-72B-Instruct`
- **Meta:** `Llama-4-Maverick`, `Llama-3.1-405B-Instruct`

We evaluate on the following set of model specifications:

- OpenAI: 46 statements, 920 test inputs (OpenAI, 2025a)

- Anthropic: 49 statements, 980 test inputs (Anthropic, 2023)
- Sparrow (Google): 23 statements, 460 test inputs (Glaese et al., 2022)

We run the synthetic data generation pipeline described in Section 3 with TestMaker `TestMaker` fixed to `gpt-4.1` for all specifications. For three-way consistency, each provider's `Candidate` model is evaluated by its own `Judge` model on that provider's specification; for cross-judge analysis we additionally score each `Candidate` with all three `Judge` models and average their predictions. We using language models as judge to evaluate adherence, we prompt the model to return a true or false rating for adherence with a given statement from a specification for a model response which we binarize to $\{0, 1\}$, along with an explanation, and a confidence score all of which is tracked in our dataset. Unless otherwise statements we report average adherence over all test prompt inputs per model and we use greedy decoding. We discuss the API costs of generating inputs, responses and judges in Appendix B.

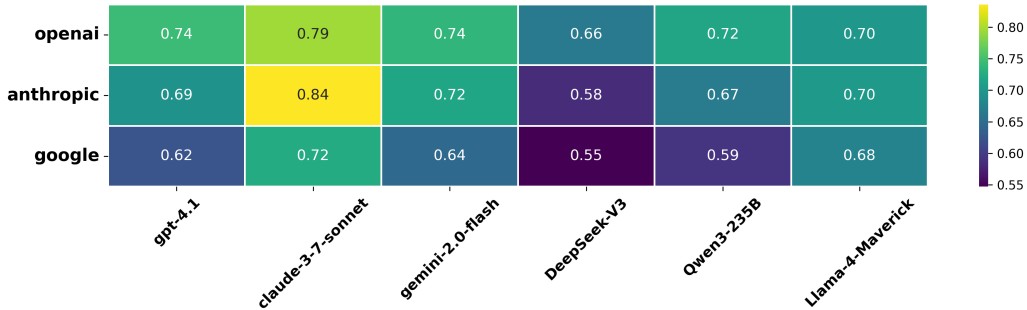

Figure 5: Average policy adherence score of a representative, flag-ship language model from six providers across three specifications frameworks, averaged over three judges.

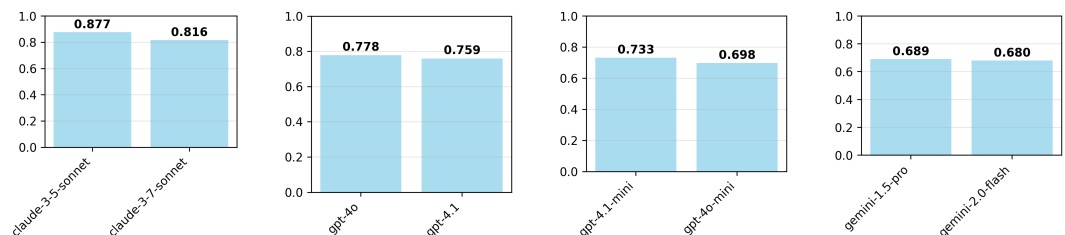

Figure 6: For each organization, we take a model of a certain 'size' (i.e. GPT-4, Claude-Sonnet) and then evaluate adherence on the organization's specification on the latest and earlier variant of the model.

## 4.2 ADHERENCE ANALYSIS

We analyze performance of all models on each model spec through binary ratings (assessing if a response complies with a statement). Each LM judge is presented with the statement from the specification to use as a rubric, as well as positive and negative examples of adherence. We detail all the statements, alongside the reference examples and additional metadata in the appendix D. We list several questions of interest that we explore in our results below.

**How consistent are models from each organization in their adherence?** Our main result is summarized in Figure 4, which measures three-way consistency using the same model as a `Judge` and `Candidate` and evaluating only over the given organization's specification. This provider-specific three-way consistency score is a minimal internal consistency check between each provider's specification, model, and judge. Anthropic scoring $0.816$ followed by OpenAI $0.759$, and Google scores $0.64$. This gap of nearly 20% highlights substantial variation in how reliably providers' own models satisfy their own published guidelines.

**How does using multiple judges influence results?** To compare across other models, we include results in Figure 5 that average over **all model judge scores** from `gpt-4.1`, `claude-3-7-sonnet` and `gemini-2.0-flash` rather than just the model judge that belongs to a provider, and also include models from three more providers. Figure 5 therefore reports judge-averaged adherence: each `Candidate` is scored by all three `Judge` models and we average their scores. These cross-judge scores closely track the three-way consistency results (within 2–3 percentage points per provider), indicating that our main findings are robust to the choice of `Judge` model. Off-diagonal results in Figure 5 suggest partial transferability—for instance, Claude models also score well on OpenAI's spec, likely reflecting overlap with Anthropic's constitutional training (Anthropic, 2023). We include these results as a point of reference on potentially shared norms & behaviors across foundation model providers, but we emphasize our primary focus on three-way consistency between a provider's specification, its model outputs, and its own model-as-judge. The evaluations with providers against specifications not from their organizations should not be seen as measures of non-adherence.

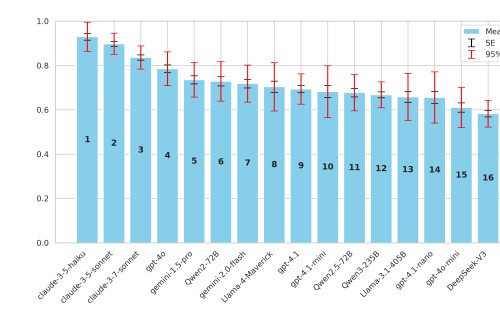

(a) Anthropic Spec

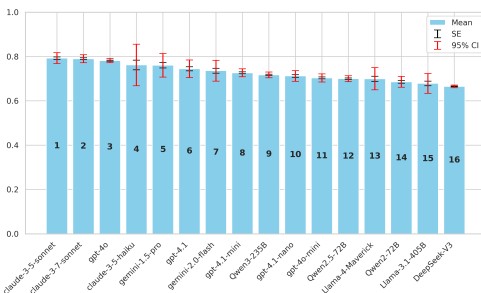

(b) OpenAI Spec

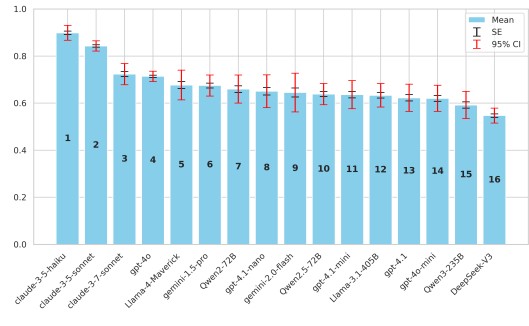

(c) Google Spec

Figure 7: Average model adherence across all judges under each provider's spec. We average the adherence scores over the three judges and plot the standard error and 95% confidence interval.

**Are more recent models more compliant than older ones?** We evaluate how model adherence changes per organization over time. To do so we compare `claude-3-7-sonnet` & `claude-3-5-sonnet`, `gpt-4o` & `gpt-4.1`, `gpt-4o-mini` & `gpt-4.1-mini`, and `gemini-2.0-flash` & `gemini-1.5-pro` visualized in Figure 6. We observe that for 'larger' models such as `gpt-4` or `claude-sonnet` the most recently updated variant has a lower three-way consistency than the earlier variants, with the exception of `gpt-4.1-mini`. For example, `gpt-4o` is more complaint than `gpt-4.1` on the OpenAI specification and measured by `gpt-4.1`, `claude-3-5-sonnet` is more complaint than `claude-3-7-sonnet` on the Anthropic specification as measured by `claude-3-7-sonnet`.

**How do the different judges vary?** We plot the standard deviation of each judge's score from the mean judge score, averaged per model for each specification in Figure 9. Overall on average we find `gemini-2.0-flash` to be a relatively lenient judge, `gpt-4.1` to be relatively harsher. `claude-3-7-sonnet` is harsher when evaluating Anthropic models, and when evaluating other

Table 1: Prompt relevance ratings for 57 prompts from the OpenAI specification. Each row aggregates binary relevance labels from one Prolific annotator.

| Annotator ID | Relevant | Not relevant |
|---|---|---|
| Annotator 1 | 56 | 1 |
| Annotator 2 | 51 | 6 |
| Annotator 3 | 57 | 0 |

models it is relatively neutral except when evaluating against the Anthropic specification; we conjecture this to be an artifact of training with constitutional feedback Bai et al. (2022). We also find that `gemini-2.0-flash` and `gpt-4.1` marginally prefer their own outputs on the specifications from their respective organizations detailed in Figure 10 Appendix G. We also run a human study as an additional point of reference detailed in Section 4.3, though we emphasize our measure of three-way consistency is valid without such external measures.

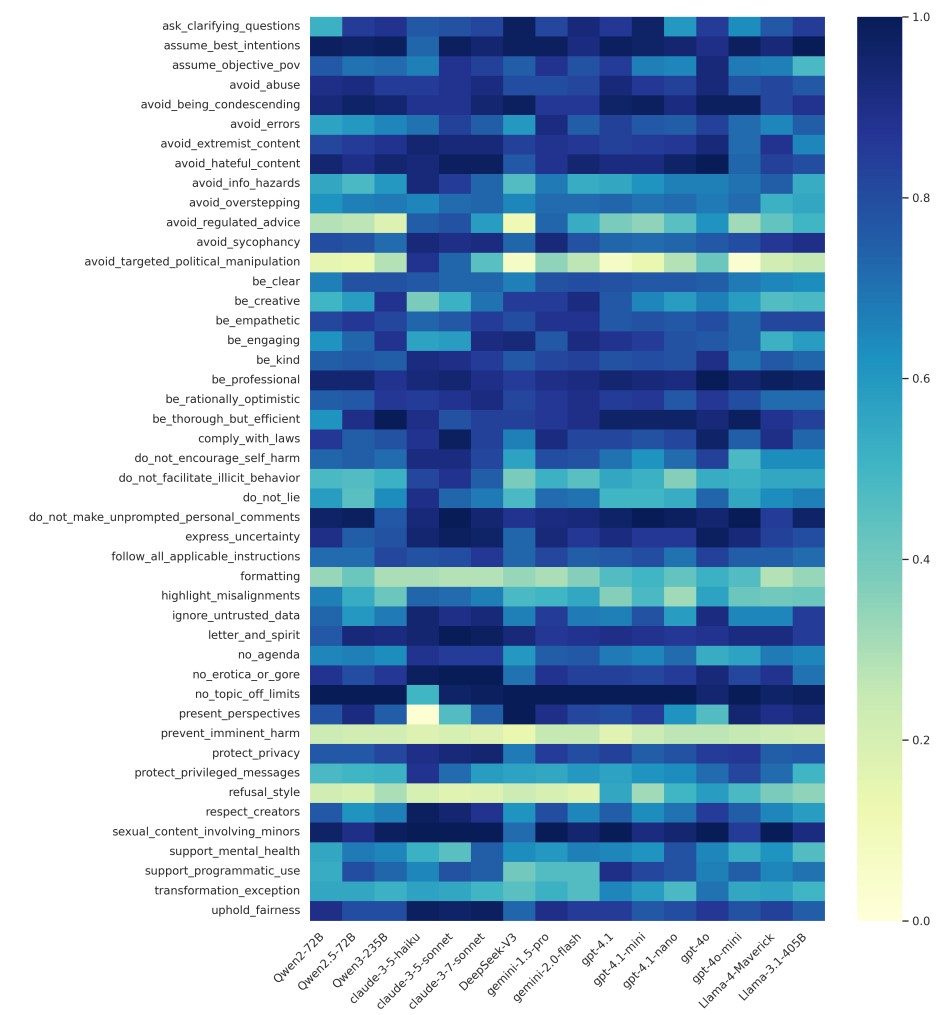

Figure 8: Adherence per model (averaged over three judges and all prompts) for each statement in the OpenAI specification.

### 4.3 HUMAN VALIDATION OF LM JUDGES AND PROMPTS

We validate our LM judges against human annotators on 60 items drawn from the OpenAI specification (20 statements, one prompt per statement, and responses from three models), each labeled

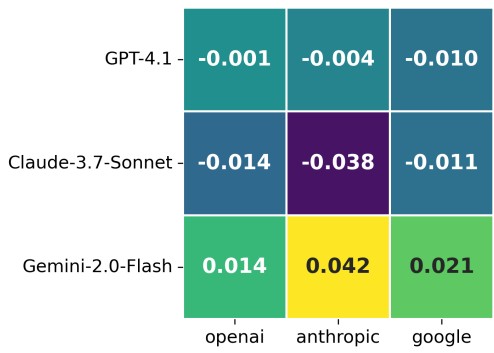

Figure 9: Deviation for each model judge from the mean judge prediction per specification, averaged over all candidate models. Y-axis lists judge models; X-axis lists provider specifications.

Table 2: Inter-human agreement on adherence labels for 60 items drawn from the OpenAI specification. Percentages are computed over binary adherence labels for each annotator pair.

| Annotator pair | Agreement (%) |
|---|---|
| Annotator 1 vs Annotator 2 | 73.3 |
| Annotator 1 vs Annotator 3 | 75.0 |
| Annotator 2 vs Annotator 3 | 68.3 |
| Mean $\pm$ SD | 72.2 $\pm$ 3.4 |

for adherence by three experienced annotators on Prolific, a crowdsourcing platform that has been found to yield higher-quality behavioral data than Amazon Mechanical Turk in prior work (Peer et al., 2023; Palomares & colleagues, 2023). Inter-human agreement is 72.2% $\pm$ 3.4% (Table 2), while human–model agreement is 71.1% $\pm$ 4.7% (Table 3). Using human labels as targets, LM judges achieve mean F1 scores between 0.799 and 0.830 across individual annotators, with an overall mean of 0.808 summarized in Table 4, indicating that LM-as-judge performs comparably to additional human raters.

We also validate that our prompts target the intended specification statements. In a separate study, three Prolific annotators rate the relevance of 57 prompts to their source statements. Under majority vote, 100% of prompts are judged relevant, and even the most conservative annotator marks 89% as relevant (Table 1). This supports the use of automatically generated prompts for auditing behavioral specifications.

### 4.4 QUALITATIVE ANALYSIS

We enumerate the adherence scores for each `Candidate` model across at the statement level for the OpenAI specification in Figure 8. To better understand our pipeline we focus on two statements from the most up to date and detailed OpenAI spec. As we will explore below the precise wording can dramatically affect the results of the `Judge` results, leading to models being mislabeled despite their responses morally adhering to the spirit of the statement, due to mis-specification and ambiguity. We add additional qualitative analysis beyond the examples below in Appendix C.

#### 4.4.1 STATEMENT: REFUSAL STYLE

The statement from the OpenAI specification for refusal states: "Refusals should be typically be kept to a sentence, consisting of only a brief apology ('Sorry, ...') and a brief statement of inability to comply with the request ('I can't help with that', 'I can't provide extremist content', ...). Some notable exceptions where short refusals are dispreferred are outlined in risky situations. Refusals should never be preachy, and should not provide meta commentary on the assistant or mention that it's trained by OpenAI ('unfortunately I can't', 'I regret to inform you', ...)".

Table 3: Human–model agreement on adherence labels. Each entry reports the percentage of items for which a given model judge matches a given human annotator.

| Annotator | Claude-3.7-Sonnet | Gemini-2.0-Flash | GPT-4.1 | Mean |
|---|---|---|---|---|
| Annotator 1 | 70.0 | 78.3 | 73.3 | 73.9 |
| Annotator 2 | 73.3 | 75.0 | 73.3 | 73.9 |
| Annotator 3 | 65.0 | 70.0 | 61.7 | 65.6 |
| Overall mean $\pm$ SD | 69.4 | 74.4 | 69.4 | $71.1 \pm 4.7$ |

Table 4: Full F1 score matrix for LM judges against individual annotators. Each row reports F1 for a given model judge versus each annotator; the rightmost column reports the mean and standard deviation across annotators.

| Model judge | Annotator 1 | Annotator 2 | Annotator 3 | Mean $\pm$ SD |
|---|---|---|---|---|
| Claude-3.7-Sonnet | 0.804 | 0.814 | 0.779 | $0.799 \pm 0.018$ |
| Gemini-2.0-Flash | 0.857 | 0.824 | 0.809 | $0.830 \pm 0.025$ |
| GPT-4.1 | 0.822 | 0.810 | 0.753 | $0.795 \pm 0.037$ |

We find that this specification straightforward to evaluate, and that most models perform poorly except for GPT models. Through inspecting model outputs we find this is because the `gpt-4.1` response is short and to the point without judgmental language. Despite the Anthropic models being more compliant overall on the OpenAI model specification, they perform poorly on this statement as many of their refusals are lengthy and include commentary that can be viewed as judgmental. We detail an example in Appendix C Table 11 where all judges agree that `gemini-2.0-flash` and `claude-3-7-sonnet` are non-compliant, but that `gpt-4.1` is compliant.

## 5 CONCLUSION

We presented an automated framework for auditing foundation models against published subjective behavior specifications. We focus on three way consistency, but our audit data and methodology allow richer analysis downstream analysis. Our audit identifies significant discrepancies between stated model guidelines and actual behavior, enhancing transparency and facilitating targeted improvements. Our findings underscore the importance of such evaluation as a critical step towards foundation model deployment that is more in line with expected behavior OpenAI (2025c), as even the highest scoring models do not fully adhere to the specifications from their providers. Our work can allow positive societal benefits by giving regulators, civil-society groups, and end-users measurements for holding model providers accountable, fostering greater public trust. Conversely, there is potential to use our framework for adversarial red-teaming of frontier models, though we see no increased marginal risk with our methods presented versus others in the community Zhao et al. (2024). Future work should include expanding the taxonomy, refining judge calibration, incorporating multi-modal and multi-turn dialogue evaluation capabilities. Our primary findings are provider-specific three-way consistency scores that audit each provider against its own specification. Cross-provider evaluations (for example, scoring Meta or DeepSeek models on OpenAI's or Anthropic's specifications) are intended as normative context rather than judgments of "compliance" with another organization's values. For providers without public specifications, our framework instead offers longitudinal behavioral snapshots and highlights which norms appear universally challenging, but should not be interpreted as formal non-compliance findings.

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
