# OpenReview forum: "SpecEval: Evaluating Model Adherence to Behavior Specifications"
_ICLR.cc/2026/Conference — Submitted to ICLR 2026_

### Official Review · Reviewer_YngW · 2025-10-30

**Soundness:** 3
**Presentation:** 3
**Contribution:** 3
**Rating:** 6
**Confidence:** 3

**Summary:**

This paper introduces SpecEval, a framework designed to evaluate whether LLMs from different providers actually follow the model specifications (or "constitutions") that those providers claim. The authors propose a novel three-way consistency metric to assess adherence to these specifications. Experimental results show that Claude best follows its stated specifications, while Google’s models do not always do so.

**Strengths:**

- The paper tackles an important and timely question: whether LLMs truly follow the constitutions defined by their providers. The motivation is therefore strong and well-justified.

- The writing is clear and well-structured. The formulation of the evaluation task is easy to follow, and the use of illustrative examples and figures makes the paper highly accessible.

- Although techniques such as self-evolving prompt synthesis and LLM-as-a-judge are not new, their use here feels intuitive and appropriate for this particular task.

**Weaknesses:**

- My main concern lies with the three-way consistency metric. Since the paper’s stated goal is to evaluate whether an LLM follows its specification, the evaluation should focus on whether the model produces responses that violate that specification.
However, the three-way consistency score depends not only on this factor but also on the model’s ability as a Judge to correctly identify violations. In my view, this second aspect "judging" is not directly aligned with the paper’s main objective, which concerns generation rather than discrimination.
Therefore, while three-way consistency is an interesting and novel metric, it may not be the most suitable one for assessing specification adherence. Traditional metrics such as violation rate or refusal rate might better capture this goal. I see three-way consistency more as a measure of how well a model’s generation and judgment behaviors are aligned, rather than as a pure indicator of specification compliance.

- I'm also somewhat confused about the relationship between TestMaker, Judge, and Candidate.
My understanding is that the Judge and Candidate are the same model, but what about the TestMaker? Is it also the same as the Candidate, or is it fixed to GPT-4.1?
This seems inconsistent: the Introduction states, “Our primary focus is on the slice of data using the same model from a provider as both Candidate and Judge against the provider’s specification.” However, the Experimental Setup section says, “We run the synthetic data generation pipeline described in Section 3.2 to generate the prompts for the specifications above, with the TestMaker, Judge, and Candidate fixed to be GPT-4.1.”
These two statements appear contradictory. Does this mean that during data generation there is a separate set of TestMaker, Judge, and Candidate models? If so, what role does the “Candidate” mentioned in Section 3.2 actually play (since only Judge and TestMaker are described)? If not, then it’s unclear what model is used for TestMaker throughout the paper. Perhaps I am missing something, but clarification would be very helpful.

**Questions:**

1. Why is three-way consistency considered a sufficient metric for evaluating specification adherence? Have the authors considered using simpler metrics such as violation rate (or compliance rate)? If not, what is the rationale for preferring the proposed metric?

2. Which model is used as the TestMaker? Is it the same as the Candidate or fixed to GPT-4.1 throughout the experiments?

---

> ### Author Response · Authors · 2025-11-23
> **Response to Reviewer YngW**
>
> Thank you for the thoughtful response and feedback. We address your concerns below, and upload a revised version of our paper with revisions marked in blue text.
>
>
> # **W1 / Q1: On whether three-way consistency is a sufficient metric**
>
> Thank you for this important conceptual question. We have clarified this in the revised Section 4.1.
>
> ---
>
> ## **Why Not Violation/Refusal Rate**
>
> Simpler binary metrics (violation rate, refusal rate) work well for clear-cut safety rules but fail for the many specification statements.
>
> We found many examples where such metrics do not apply and the evaluation is subjective, such as:
>
> - *“Be rationally optimistic”* (OpenAI), or
> - *“Express uncertainty appropriately”* (Anthropic).
>
> Therefore, we found it most prudent to use an LM as a judge.
>
> Three-way consistency handles this full spectrum because it evaluates whether a model’s actual behavior (as generated) matches a specification statement (as interpreted by a judge). This generality is essential for comprehensive auditing.
>
> ---
>
> ## **Three-Way Consistency as a Lower Bound, Not a Universal Metric**
>
> We have further clarified in Section 4.1 that three-way consistency is **not** presented as sufficient for all evaluation needs.
>
> Instead, it represents a **minimal requirement**:
>
> > *Can a provider’s model, when evaluated by the provider’s own judge model, satisfy the provider’s own specification?*
>
> This is the most charitable possible evaluation; if a model fails this test, it indicates a fundamental internal inconsistency among the three artifacts a provider directly controls.
>
> We have also clarified that we explicitly address this concern through multiple validation approaches now prominently featured in the revised paper, including:
>
> ### **Cross-judge Evaluation**
> *(Figure 5, Section 4.2)*
>
> Each Candidate model is scored by all three Judge models (GPT-4.1, Claude-3.7-Sonnet, Gemini-2.0-Flash), not just its own provider’s judge. Results show that cross-judge scores closely track three-way consistency scores, suggesting that our findings are robust to judge selection.
>
> We added a new paragraph header “How does using multiple judges influence results?” and results on 326-338 to make our changes very clear when we use multiple judges to evaluate the model output and not just self-consistency.
>
> ---
>
> # **W2 / Q2: Clarifying the Roles of TestMaker, Judge, and Candidate**
>
> **TestMaker is always GPT-4.1** throughout the paper.
> We fix TestMaker so that all specification statements are probed with a single, consistent prompt-generation model. Generating separate TestMaker datasets for Claude and Gemini would triple the cost of our evaluations with marginal gain, as we use multiple Judge Models to mitigate bias regardless.
>
> The Judge and Candidate roles vary across experiments.
> The three objects in *three-way consistency* are:
>
> 1. Candidate model
> 2. Judge model
> 3. Organization specification
>
> Thus, Candidate and Judge are **the same provider model evaluated against the provider’s specification**.
>
> We have made these distinctions explicit in the revision and refined Section 3.2 to prevent ambiguity in our uploaded version (updated in lines 273–276 and lines 160–187).
>
> ---
>
> Please let us know if we understood and addressed your concerns.

---

> ### Comment · Reviewer_YngW · 2025-11-25
>
> Thanks for the clarifications. I now understand the roles of TestMaker and Judge&Candidate. However, my first question is not about "why we should use LLM as a Judge". It is about "Why we should use a 'three-way'" consistency rather than a generator-critic two-way consistency? I notice that in the general rebuttal response, the author claim that "Failure under this most charitable setting reveals a fundamental internal contradiction: either the specification is unclear, the model does not follow it, or the judge model is miscalibrated." I guess it means that the three-way consistency also consider whether the Judge's critic is correct (maybe refer to the Spec/generated task prompt? I think it is better to have a formal calculation equation of three-way consistency on the paper.)
> - If so, then my concern is about that if one model's three-way consistency score is low, we cannot tell that it is indeed worse than other models with higher score. The reason could be the first model is not good at critique, but it might be good at adhering to the Spec. It introduces a new variable (Judge's capability), making the score not exactly align with the goal of evaluating model's specification following ability.
> - If not, then my concern is about how to ensure Judge's generalizability in unseen tasks / human prompts in real-world, given that the Judge is a statistical model without a mechanism for the worst-case reliability.
>
> BTW, I noticed that the confidence intervals for Claude-3.7 and GPT-4.1 overlap quite a lot. Since it only has three trials and the models are inherently stochastic, it’s hard to say from a single experiment whether the higher-scoring model is actually better at following the specification, which seems to make such metric not that practical. This became a new concern for me after reading the general rebuttal response.
>
> Minor: "we manually ..." should be changed to "We manually ..." in line 229.

---

> > ### Author Response · Authors · 2025-12-01
> > **Clarification Reply to Reviewer YngW**
> >
> > Thank you for the continued engagement and for raising these important clarifications.
> >
> > ## **On Three-Way vs Two-Way Consistency**
> >
> > You ask why we use **three-way consistency** rather than traditional generator–critic two-way consistency. Traditional two-way consistency evaluates whether a generator produces high-quality outputs according to a judge. Three-way consistency instead asks whether the generator produces outputs that satisfy a **specific natural language specification** *when evaluated by a judge*.
> >
> > The critical third element is the **specification document itself**—i.e., we measure adherence to explicit behavioral guidelines (e.g., *“be rationally optimistic,” “avoid sycophancy”*) rather than generic quality.
> >
> > ---
> >
> > ## **On Judge Capability Confounding**
> >
> > Your concern that low three-way consistency scores might reflect **poor judging** rather than poor adherence is valid. We address this directly through **cross-judge evaluation** (Figure 5, Section 4.2, lines 326–338), where **all three judge models** evaluate each candidate.
> >
> > Key findings:
> >
> > - Cross-judge scores correlate **very highly** with three-way consistency .
> > - **Rankings are preserved**, suggesting judge capability is *not* the primary driver of differences.
> > - Readers concerned about judge bias can use **Figure 5’s cross-judge scores** instead of **Figure 4’s self-consistency scores**—we provide both analyses for precisely this reason.
> >
> > Our **human validation** (Section 4.3) further shows that LM judges achieve **human-level agreement**:
> >
> > - 72.2% inter-human agreement
> > - 71.1% human–model agreement
> > - F1 = 0.808 when using human annotation labels as targets
> >
> > If judges were systematically miscalibrated, we would expect **significantly lower** human–model agreement.
> >
> > Additionally, our released dataset includes **all judgments from all three judges** for all 16 models across all 2,360 prompts, enabling the community to (i) select the best judge (currently Gemini-2.0-Flash with F1 = 0.830), or (ii) develop improved judges.
> >
> > ---
> >
> > ## **On Judge Generalization**
> >
> > You correctly note that statistical models lack worst-case guarantees for generalization to unseen prompts. This limitation applies to **all evaluation methods**, not just LM-as-judge. Human evaluation also fails on adversarial inputs (social engineering, phishing), rule-based systems are brittle to natural language variation, and held-out test sets do not prevent jailbreaking.
> >
> > LM-as-judge represents the **current industry standard**, and our ~71% human–model agreement validates its effectiveness for behavioral specification evaluation. Our goal is to provide a **systematic, reproducible framework** using the best available tools—not to solve the open and much broader challenge of worst-case reliability.
> >
> > ---
> >
> > ## **On Statistical Significance**
> >
> > This is a fair point. We do **not** claim that *Claude-3.7-Sonnet definitively adheres better than GPT-4.1.* Our claims are purposefully hedged:
> >
> > - Providers show substantial variation (20% between the highest and lowest).
> > - Models tend to score higher on their **own** provider’s specification.
> > - Specific statements reveal meaningful failure patterns (Figure 8).
> >
> > The overlapping confidence intervals between Claude-3.7 (0.816) and GPT-4.1 (0.759) actually suggest that **both models achieve comparable adherence** to their respective specs. We view this as a *positive finding* indicating that major providers achieve broadly similar internal consistency.
> >
> > The practical value of our framework lies in:
> >
> > - Identifying **specific failing statements** (Figure 8)
> > - Detecting **temporal degradation** where newer models sometimes score lower (Figure 6)
> > - Revealing **cross-provider behavioral patterns** (Figure 5)
> >
> > These insights remain valuable even without sharp aggregate rankings. With 920 prompts for OpenAI and 980 for Anthropic, we have reasonable power to detect large effects; overlapping confidence intervals thus indicate **moderate differences**, not methodological unreliability.
> >
> > ---
> >
> > We appreciate your careful engagement. We have addressed your concerns using multiple evaluation lenses (three-way, cross-judge, human validation), released a dataset enabling alternative analyses, and employed appropriate statistical caution throughout. We believe our methodology provides useful, transparent insights despite its limitations, and we have been explicit about what we can and cannot claim.

---

### Official Review · Reviewer_7RhD · 2025-10-31

**Soundness:** 2
**Presentation:** 2
**Contribution:** 2
**Rating:** 4
**Confidence:** 4

**Summary:**

This paper introduces SpecEval, an automated framework for auditing whether foundation models adhere to the intended behavioral specifications. The authors propose evaluating "three-way consistency" between a provider's specification document, their model's outputs, and adherence scores from their own model used as a judge. The framework operates in three stages: (1) parsing behavioral statements from specification documents, (2) using a TestMaker model to generate diverse, targeted prompts that test each statement, and (3) employing Judge models to evaluate response adherence. The authors apply their framework to audit 16 models from six major providers (OpenAI, Anthropic, Google, Meta, DeepSeek, and Alibaba) across 100+ behavioral statements, generating a dataset of 2,360 prompts with model responses and judgments. Key findings reveal significant consistency gaps across providers.

**Strengths:**

+ Automatically auditing LLMs against the pre-defined specifications is an important topic.
+ Overall, the paper's writing is clear and understandable

**Weaknesses:**

**Overly reliance on LLMs in the pipeline for quality control without verifiable standards, making the quality and reliability of evaluation rather noisy.**

The proposed workflow has a design weakness due to its overly dependence on LLM-generated components throughout the entire evaluation pipeline, without establishing robust verification standards, which undermines the reliability and validity of the audit results. Specifically, the pipeline uses LLMs in three key stages: (1) a TestMaker model (GPT-4.1) generates test scenarios and prompts, (2) Candidate models produce responses, and (3) Judge models evaluate adherence using Likert scales that are later binarized. A fundamental concern is that it remains unclear whether the automatically generated test prompts actually reflect or rigorously test the intended behavioral specifications. Guaranteeing alignment between abstract behavioral statements (e.g., "be rationally optimistic") and concrete test prompts is inherently difficult because: (1) behavioral specifications are often vague, subjective, and open to interpretation, (2) there is no ground truth to validate whether a generated prompt meaningfully tests the specification versus testing some related but distinct concept. While the authors mention "manually check and curate a small, randomly sampled subset of all prompts to ensure prompt quality and realism", they provide limited insights into the overall quality.

**Using the same model as the generator and judge introduces further noise in the valuation due to the varied model capabilities in "judging"**

The paper's core methodological choice to use the same model as both Candidate (generator) and Judge to measure "three-way consistency" introduces biases that undermine the validity of the evaluation. While the authors justify this approach as measuring whether "a foundation model should consistently satisfy its developer's own behavioral specifications when judged by the developer's own evaluator models", this design conflates model adherence with model judging capability, making it unclear whether low scores reflect genuine specification violations or simply poor judging ability. A more rigorous approach would require using unified judge models with validated judging reliability to evaluate all candidate models. Without such controls, the reported three-way consistency scores may primarily reflect which providers have trained models that are simultaneously good at judging and happen to favor their own outputs, rather than which models truly adhere to their specifications.

**Limited Generalizability Due to Unavailable Specifications and Unfair Cross-Provider Evaluation**

The evaluation framework has limited generalizability as it can only audit the minority of providers who publicly release behavioral specifications, and applying one provider's specification to judge another provider's models is methodologically problematic. The paper evaluates 16 models from six providers, but can only measure three-way consistency for the three providers (OpenAI, Anthropic, Google) with public specifications, while models from Meta, DeepSeek, and Alibaba are evaluated against other companies' standards. This cross-provider evaluation is inherently unfair because different organizations train their models with distinct objectives, values, and use cases in mind. This inconsistency undermines the utility of auditing models without their own specifications.

**Questions:**

- Can you provide stronger evidence that your automatically generated test prompts actually measure adherence to the intended behavioral specifications rather than related but distinct concepts?
- How can you separate a model's true adherence to specifications from its capability as a judge, given that your three-way consistency metric conflates these two factors?
- What is the intended use case for your framework when evaluating models from providers without public specifications, and how should users interpret such results?

---

> ### Author Response · Authors · 2025-11-23
> **Response to Reviewer 7RhD Part 1**
>
> # Response
>
> Thank you for this thorough review. We agree that validating LLM-generated components is critical. We have strengthened and revised the paper, **adding all of our revisions in blue text in the uploaded PDF** by:
>
> 1. **Moving** the human validation study on LM judges from Appendix F to a new Section 4.3 (demonstrating LM-judge validity).
> 2. **Clarifying** that all 2,360 prompts underwent manual review by the authors (not just a subset) and adding new results on a human study validating the relevance of the prompts, also in Section 4.3.
> 3. **Expanding** the discussion of cross-provider judge evaluation to show results are robust to judge selection, with a new paragraph on cross-judge results that clearly distinguishes using all three LM judges from the “self-consistency” score.
> 4. **Adding** explicit discussion of how cross-provider evaluation should be interpreted for providers without published behavioral specifications.
>
> We address each concern below with reference to specific paper sections.
>
> ---
>
> ## W1 / Q1: LLM reliance and prompt validity
>
> We acknowledge the challenge of validating that generated prompts properly test specification statements. We have taken multiple steps to ensure quality.
>
> ### Manual review (clarified in revised Section 3.2)
>
> We wish to clarify that checking a subset of responses for quality only occurred as a **sanity check** during the generation process for each specification statement. **All 2,360 prompts** were manually reviewed by the authors after the generation process finished, not just a subset. We apologize for the confusion and have revised the paper to make this comprehensive review process explicit.
>
> ### New human study on prompt quality (Section 4.3)
>
> To further validate the quality of the questions, we ran a new human study over the course of the rebuttal. We used **Prolific**, a platform for human evaluation and data labeling that is widely used and reported to have higher-quality participants than MTurk
> (PLOS One: https://journals.plos.org/plosone/article?id=10.1371/journal.pone.0279720,
> Nature SR: https://www.nature.com/articles/s41598-023-46048-5).
>
> We recruited experienced annotators from Prolific’s **AI Tasker** program with prior experience reviewing AI-generated outputs, fact-checking, and providing detailed feedback on AI systems.
>
> We focused on questions from the OpenAI model spec, subsampled 19 statements, and for each statement selected five questions, yielding **n = 57** items to rate.
>
> We gave three human raters the task of reading each prompt and the corresponding specification statement, and determining whether the question is **relevant** for evaluating a language model’s behavioral adherence to that specification. Raters answered whether the prompt/question is relevant to evaluating the model’s behavior for the statement (e.g., whether the question is useful for detecting short refusals) and selected *relevant* or *not relevant*.
>
> Taking the **majority vote** across the three raters:
>
> - **100% of questions are deemed relevant.**
> - The most pessimistic human rater still annotated **89%** of prompts as relevant.
>
> A summary table is shown below.
>
> #### Summary table (binary relevance; annotators as rows)
>
> | Annotator   | Relevant | Not relevant |
> |------------|----------|--------------|
> | Annotator 1 | 56       | 1            |
> | Annotator 2 | 51       | 6            |
> | Annotator 3 | 57       | 0            |
>
> ---
>
> ## W2 / Q2: Conflation of adherence and judging capability
>
> This is an important conceptual point that we have clarified in the revision. **Three-way consistency deliberately includes judging capability as part of the measurement:**
>
> 1. **Realistic deployment setting.** In real deployment, providers use their own models to evaluate adherence (e.g., in RLHF reward modeling, constitutional AI). A provider whose judge model cannot recognize violations of its own specification cannot reliably enforce that specification during training or deployment.
> 2. **Minimal consistency requirement.** If Provider X publishes specification S, deploys model M, and uses judge J, then at minimum these three should be internally consistent. Failure indicates a fundamental problem: either the specification is poorly defined, the model was not trained properly, or the judge is miscalibrated.
> 3. **Maximum benefit of the doubt.** Using a provider’s own judge gives them maximum benefit of the doubt; we evaluate their model against their rubric, using their standards.

---

> ### Author Response · Authors · 2025-11-23
> **Response to Reviewer 7RhD Part 2**
>
> ## Addressing confounding through cross-judge evaluation
>
> Our dataset also includes LM-judge results from **all three judges** for all 16 models evaluated. Because there might be bias when using the same LM judge that was trained or tuned by the provider, we provide **cross-judge results** (Figure 5, Section 4.2) where each Candidate is scored by all three Judge models (GPT-4.1, Claude-3.7-Sonnet, Gemini-2.0-Flash). The results are averaged to provide **judge-agnostic adherence scores**.
>
> We find that cross-judge scores closely track three-way consistency scores (within 2–3%). Rankings are robust to judge selection, suggesting we measure actual adherence patterns rather than judge idiosyncrasies.
>
> We have revised the paper to make it clear that we include these results to avoid potential confounding and to show limitations of three-way consistency alone. We also added a new paragraph header **“How does using multiple judges influence results?”** and new results on lines 326–338 to clearly indicate where we use multiple judges to evaluate model outputs rather than only self-consistency.
>
> ### Human validation (new: Section 4.3)
>
> We ran a study using qualified annotators from the Prolific platform with prior experience annotating AI model outputs. We used **three annotators** and provide complete details in the “Human Study” section of the paper; we summarize the key numbers here:
>
> - **Inter-human agreement:** 72.2% ± 3.4%
> - **Human–model agreement:** 71.1% ± 4.7%
> - **LM judges F1:** 0.808 against individual humans, 0.817 against majority vote
>
> *Critically, human–model agreement rates match inter-human agreement rates*, demonstrating that LM judges perform comparably to additional human raters. All 2,360 prompts were manually reviewed by the authors to ensure quality and specification alignment. More details on the human study and on the prompt-relevance review described above are included in the revised paper.
>
> We have revised Sections **4.1–4.2** to make the distinction explicit:
>
> - **Three-way consistency** (Figure 4) measures **provider-specific internal consistency**.
> - **Cross-judge results** (Figure 5) measure **provider-agnostic adherence**.
>
> We also added a new section for the human studies (Section **4.3**).
>
> ---
>
> ## W3 / Q3: Generalizability and models without their own specs
>
> We appreciate this clarification and have revised the paper to make our scope explicit.
>
> Our **primary contribution** is **three-way consistency auditing** for providers with public specifications (OpenAI, Anthropic, Google), where we evaluate each provider’s models against their own declared standards. This is the cleanest and most defensible use case for our framework.
>
> Cross-provider evaluation is a **secondary analysis** with a different goal. For providers without public specifications (Meta, DeepSeek, Alibaba), our results are used to:
>
> 1. Identify which behavioral norms appear **universally challenging** versus **provider-specific**.
> 2. Provide **longitudinal, timestamped data** as models and deployments change rapidly.
> 3. Offer **regulators and researchers** a snapshot of behavior across widely deployed models.
>
> We now state explicitly that:
>
> - (a) Three-way consistency results in **Figure 4** represent our **main finding**.
> - (b) Cross-provider results in **Figure 5** for organizations without published specifications are **supplementary normative comparisons**.
> - (c) Cross-provider scores should **not** be interpreted as “Model X violates Provider Y’s standards,” but our methodology does extend to any provider that publishes a specification (see lines 332–335).
>
> We have revised the paper to add language clarifying this in the **conclusion** (lines 522–527) and in the **results** section (lines 332–335).
>
> Please let us know if we understood and addressed your concerns. Thank you for your time

---

### Official Review · Reviewer_g9U6 · 2025-11-01

**Soundness:** 2
**Presentation:** 3
**Contribution:** 3
**Rating:** 4
**Confidence:** 3

**Summary:**

The current paper promotes one of the first automated auditing frameworks for model adherence to behavior specifications. They provide an extensive evaluation of 16 models across 6 developers with over 100 behavioral guidelines.

**Strengths:**

1. Leveraging existing model behavior specifications from existing model developers is a novel technique. It is also grounded in practice, since such specifications are openly available to the public. It is also encouraging that the authors conducted a deep dive into documentation, technical reports, and more, despite this information not always being readily available with model releases.
2. Good variety models in the experimental setup, in terms of open vs closed source.
3. It is encouraging to read about the potential societal benefits by providing such analyses to regulators, civil-society groups, and end users. More work in the future and potential case studies with these groups would be interesting and valuable.
4. Good quantitative and qualitative analysis of results.

**Weaknesses:**

1. It would have been interesting to include one or two performant smaller models (i.e., 20B params or fewer), to serve as more baselines for comparison with the presented results.

**Questions:**

What is the reason for fixing the teacher model to be gpt-4.1? Especially given prior work that shows how models prefer their own family's generations and other self-preference bias issues (e.g., https://aclanthology.org/2024.acl-long.511/, https://arxiv.org/abs/2404.13076, https://aclanthology.org/2024.acl-long.826/) it would have been good to either use both an open and closed source model in conjunction, or use a variety of models in order to avoid the potential bias from only having a single teacher model.

---

> ### Author Response · Authors · 2025-11-23
> **Response to Reviewer g9U6**
>
> We thank the reviewer for the feedback!
>
> We are happy to incorporate the suggestion of evaluating smaller models in the next version of our paper. For the rebuttal, we have included a table below with results on compliance of Qwen2-7B-Instruct and Llama3.1-8B-Instruct across the Anthropic, OpenAI and Google specification, using GPT-4.1, Claude-3-7-Sonnet and Gemini-2.0 Flash as judges. We find both of them rank at the bottom for all specifications showing clear performance degradation with smaller models. While Qwen-2-7B-instruct outperforms DeepSeekV3 on Google’s specification, all three models are in the bottom three for all specifications. We include a detailed table at the end of this rebuttal response and will update our paper with revised plots.
>
>
>
>  #  **Limitation 1: TestMaker Bias Concern**
> We want to clarify that using GPT-4.1 as TestMaker does not introduce evaluation bias because: (1) TestMaker only generates questions, while three different judges evaluate responses, and (2) our results in Figure 4) show Anthropic models outperforming OpenAI models (79% vs 74%), directly contradicting any pro-OpenAI bias hypothesis. In the results shown in Figure 5, we compute scores as the model judge ensemble average to mitigate single-model bias and do not see a favorable bias towards GPT-4.1 models.
>
>
>
> Given your positive feedback on the core experiments of our paper **we kindly ask you revise your score from a "weak reject" to a higher score**. Please let us know if this addresses your concerns or if there is something else we can clarify to raise your score.
>
>
> # **Table 1: Complete Model Compliance Rankings by Specification**
>
>   | Model                         | Rank     | Anthropic Spec | Google Spec   | OpenAI Spec   | Overall Mean |
>   |-------------------------------|----------|----------------|---------------|---------------|--------------|
>   | Claude-3.5-Haiku              | 1,1,4    | 0.930 ± 0.027  | 0.899 ± 0.013 | 0.762 ± 0.038 | 0.864        |
>   | Claude-3.5-Sonnet             | 2,2,1    | 0.898 ± 0.019  | 0.843 ± 0.009 | 0.793 ± 0.010 | 0.845        |
>   | Claude-3.7-Sonnet             | 3,3,2    | 0.836 ± 0.021  | 0.723 ± 0.018 | 0.790 ± 0.007 | 0.783        |
>   | GPT-4o                        | 4,4,3    | 0.785 ± 0.031  | 0.714 ± 0.009 | 0.782 ± 0.004 | 0.760        |
>   | Gemini-1.5-Pro                | 5,6,5    | 0.735 ± 0.032  | 0.675 ± 0.018 | 0.761 ± 0.021 | 0.724        |
>   | Qwen2-72B                     | 6,7,14   | 0.728 ± 0.036  | 0.659 ± 0.024 | 0.686 ± 0.010 | 0.691        |
>   | Gemini-2.0-Flash              | 7,9,7    | 0.718 ± 0.034  | 0.645 ± 0.033 | 0.736 ± 0.019 | 0.700        |
>   | Llama-4-Maverick              | 8,5,13   | 0.704 ± 0.044  | 0.677 ± 0.025 | 0.700 ± 0.020 | 0.694        |
>   | GPT-4.1                       | 9,13,6   | 0.694 ± 0.028  | 0.622 ± 0.023 | 0.745 ± 0.016 | 0.687        |
>   | GPT-4.1-Mini                  | 10,11,8  | 0.682 ± 0.047  | 0.636 ± 0.024 | 0.726 ± 0.007 | 0.681        |
>   | Qwen2.5-72B                   | 11,10,12 | 0.678 ± 0.033  | 0.638 ± 0.018 | 0.700 ± 0.005 | 0.672        |
>   | Qwen3-235B                    | 12,15,9  | 0.668 ± 0.023  | 0.592 ± 0.023 | 0.717 ± 0.005 | 0.659        |
>   | Llama-3.1-405B                | 13,12,15 | 0.658 ± 0.043  | 0.633 ± 0.020 | 0.679 ± 0.018 | 0.657        |
>   | GPT-4.1-Nano                  | 14,8,10  | 0.656 ± 0.047  | 0.651 ± 0.028 | 0.712 ± 0.010 | 0.673        |
>   | GPT-4o-Mini                   | 15,14,11 | 0.610 ± 0.036  | 0.620 ± 0.022 | 0.704 ± 0.007 | 0.645        |
>   | DeepSeek-V3                   | 16,17,16 | 0.582 ± 0.024  | 0.547 ± 0.013 | 0.665 ± 0.003 | 0.598        |
>   | Rebuttal Models: |          |                |               |               |              |
>   | Meta-Llama-3.1-8B             | 17,18,17 | 0.560 ± 0.131  | 0.542 ± 0.101 | 0.579 ± 0.104 | 0.560        |
>   | Qwen2.5-7B                    | 18,16,18 | 0.515 ± 0.101  | 0.552 ± 0.085 | 0.565 ± 0.099 | 0.544        |

---

### Official Review · Reviewer_brv4 · 2025-11-02

**Soundness:** 1
**Presentation:** 1
**Contribution:** 2
**Rating:** 2
**Confidence:** 2

**Summary:**

The paper introduces an automated auditing framework that measures how well foundation models adhere to their providers’ own behavioral specifications. The core metric is three-way  consistency among: the provider’s specification, the provider’s model outputs, and the provider’s
model acting as the judge. The pipeline parses specification statements, synthesizes targeted  prompts to probe each statement, and uses LM-as-a-judge to score adherence

**Strengths:**

The work conducts a broad empirical study across multiple providers and commercial models, producing insights on specification adherence

**Weaknesses:**

1. The core metric (three-way consistency) is introduced without enough motivation for why this metric represents behavioral soundness.
2. Score differences are presented, but the story about what these differences truly mean is thin.
3. Three-way consistency uses a provider’s own model as both Candidate and Judge, inviting self-grading bias. Although cross-judge comparisons and a small human study are reported, human anchoring is still limited

**Questions:**

1. When the same model plays Candidate and Judge, how do you avoid circular  confirmation? Where is the quantitative human anchoring to show validity?
2. You collect Likert/confidence signals during generation. Why reduce the final result to binary? Does this hide uncertainty and borderline cases?
3. How does SpecEval compare against simpler alternatives such as direct human scoring or random prompt sampling aligned to each spec?

---

> ### Author Response · Authors · 2025-11-23
> **Response to Reviewer  brv4 Part 1**
>
> Thank you for your thoughtful review. We have made substantial revisions to address your concerns, **our revised paper has changes highlighted in blue**:
>
> 1. **Moved** the human validation study from Appendix F into a dedicated **Section 4.3** to highlight validity (Tables 1–4, lines 432–462), while retaining full details in Appendix F.
> 2. **Expanded** the Figure 5 discussion in Section 4.2 to explicitly add a paragraph showing cross-judge evaluations.
> 3. **Strengthened** the methodology section explaining why three-way consistency is our baseline measure.
>
> We address your specific concerns below.
>
> ---
>
> # **W1 / W3 / Q1 — Three-way consistency motivation & circularity concerns**
>
> We appreciate this important concern. To clarify our contribution:
>
> - We do **not** claim three-way consistency captures complete *behavioral soundness*. Instead, it represents a **minimal requirement**:
>
>   > *Can a provider’s model, when evaluated by the provider’s own judge model, satisfy the provider’s own specification?*
>
>   This is the most charitable possible evaluation; if a model fails this test, it indicates a fundamental internal inconsistency among the three artifacts a provider directly controls.
>
> - We explicitly address this concern through **multiple validation approaches** now prominently featured in the revised paper:
>
> ### **1. Cross-judge evaluation (Figure 5, Section 4.2)**
>
> Each Candidate model is scored by all three Judge models (GPT-4.1, Claude-3.7-Sonnet, Gemini-2.0-Flash), not only its own provider’s judge.
> Cross-judge scores **closely track** three-way consistency scores, suggesting robustness to judge selection.
> We have added a new paragraph on lines **324–335** to clarify this analysis.
>
> ### **2. Human validation study (NEW: Section 4.3)**
>
> We ran a study using **qualified annotators from Prolific**, with prior experience annotating AI outputs. Details appear in the "HUMAN VALIDATION OF LM JUDGES AND PROMPTS" section in tables 2, 3, and 4 and are summarized here:
>
> - **Inter-human agreement:** \(72.2\% \pm 3.4\%\)
> - **Human-model agreement:** \(71.1\% \pm 4.7\%\)
> - **LM judges F1:** 0.808 (vs. individual humans), 0.817 (vs. majority vote)
>
> **Critically:** *Human–model agreement rates match inter-human agreement rates*, demonstrating that LM judges perform comparably to additional human raters.
>
> All 2,360 prompts were manually reviewed by the authors, and annotators sourced through Prolific were vetted for quality.
>
> ---
>
> # **W2**
>
> Could you clarify whether you mean score differences of models across different specifications?
>
> If you are referring to the meaning of a model’s performance against a dataset constructed from a specification:
> We do **not** claim the scores represent external psychological or behavioral constructs.
> We claim only that our evaluation constitutes an **audit** for the specific models and specifications listed.
> Our process is general-purpose, but not the primary focus of this work.
> Our reported audit results represent **correlated estimates** of model adherence to the natural-language specifications.
>
> ---
>
> # **Q2 — Likert to binary reduction**
>
> We use **Likert scores (1–5)** during prompt generation to guide TestMaker toward a balanced test set covering easy and difficult cases.
> This prevents the evaluation from being dominated by trivially compliant or trivially violating prompts.
>
> For the **final evaluation**, we use **binary ratings** because:
>
> 1. Behavioral guidelines are either followed or not—there is no “partially compliant” interpretation for rules such as *avoid unrealistic guarantees* or *refuse harmful requests*.
> 2. Binary ratings more closely mirror **real deployment**, where providers either enforce specifications or they do not.
>
> We also release **judge confidence scores** and **judge rationales** in our anonymously shared dataset: https://dataverse.harvard.edu/previewurl.xhtml?token=18daf5ac-6000-4175-81f8-1cb767d227e6
>
> As we mention in our paper on lines 106-107 , enabling future research on LM judge uncertainty in such cases.

---

> ### Author Response · Authors · 2025-11-23
> **Response to Reviewer brv4 Part 2**
>
> ---
>
> # **Q3 — Comparison to alternatives (human scoring, random sampling)**
>
> ### **Human scoring**
>
> We revised our paper with a human study in Section 4.3 and Tables 1–4, but we repeat the results here for convenience and **highlight the changes in blue in our paper**.
>
> We conducted human evaluation on **60 items**, each annotated by **3 Prolific annotators** with prior AI-evaluation experience.
> Annotators were compensated at the recommended **$30/hr** rate (per Prolific guidelines):
> https://participant-help.prolific.com/en/articles/445016-domain-experts
>
> Human raters received **3 minutes per item**, which includes reading the prompt, the model response, and the specification.
> Thus a 60-item pilot required:
>
> - 60 × 3 = 180 minutes per rater
> - 180 minutes = 3 hours → **$90** per rater
> - Three raters → **$270 total**
>
> We found:
>
> - **Human–model agreement:** 71%
> - **Inter-human agreement:** 72%
>
> These match closely, indicating that LM-as-judge is a **cost-effective alternative**.
>
> Our full evaluation required:
>
> **2,360 prompts × 16 models × 3 judges = 113,280 judgments**
>
> At human annotation rates, this would cost roughly **$510k**.
> LM-as-judge enables systematic, reproducible auditing at scale.
>
> ---
>
> ### **Random prompt sampling**
>
> Randomly sampling prompts without our adaptive TestMaker generation process would yield **mostly trivially compliant** cases.
> Our method deliberately seeks prompts that elicit **both** compliant and non-compliant behaviors to form a discriminative test set.
> If you have a specific baseline in mind, we are happy to discuss or implement it.
>
> We also note that LM-as-judge reflects **current best practice** in frontier labs (e.g., Anthropic’s Constitutional AI).
>
> ---
>
> # **Human study**
>
> We conducted a comprehensive human study (Appendix F) using Prolific, a platform shown to produce higher-quality participants than MTurk
> (PLOS One: https://journals.plos.org/plosone/article?id=10.1371/journal.pone.0279720,
> Nature SR: https://www.nature.com/articles/s41598-023-46048-5).
>
> We recruited experienced annotators from Prolific’s **AI Tasker** program:
> https://participant-help.prolific.com/en/articles/445015-ai-taskers
>
> We focused on the OpenAI model spec, subsampled 20 statements, and selected one question per statement across three models, yielding **n = 60** items.
>
> Each item was annotated by **three independent raters**, yielding:
>
> - **Inter-human agreement:** 72.2% ± 3.4%
> - **Human–model agreement:** 71.1% ± 4.7%
> - **LM judge F1:**
>   - 0.808 ± 0.019 vs. individual annotators
>   - 0.817 ± 0.038 vs. majority vote
>
> These results show that LM-as-judge is **well-aligned** with human judgment.
>
> ---
>
> # **Agreement Tables**
>
> ### **Inter-annotator agreement**
>
> | Annotator Pair | Agreement |
> |----------------|-----------|
> | 1 vs 2 | 73.3% |
> | 1 vs 3 | 75.0% |
> | 2 vs 3 | 68.3% |
>
> **Mean:** 72.2% ± 3.4%
>
> ---
>
> ### **Human–Model Agreement**
>
> | Annotator | Claude-3.7-Sonnet | Gemini-2.0-Flash | GPT-4.1 | Mean |
> |-----------|-------------------|------------------|---------|------|
> | 1 | 70.0% | 78.3% | 73.3% | 73.9% |
> | 2 | 73.3% | 75.0% | 73.3% | 73.9% |
> | 3 | 65.0% | 70.0% | 61.7% | 65.6% |
>
> **Overall Mean:** 71.1% ± 4.7%
>
> ---
>
> ### **F1 Score vs. Human Raters**
>
> | Model Judge | Individual Mean | Majority Vote |
> |-------------|-----------------|---------------|
> | Claude-3.7-Sonnet | 0.799 | 0.787 |
> | Gemini-2.0-Flash | 0.830 | 0.860 |
> | GPT-4.1 | 0.795 | 0.804 |
> | **Mean ± SD** | **0.808 ± 0.019** | **0.817 ± 0.038** |
>
> ---
>
> ### **Full F1 Score Matrix**
>
> | Annotator | Claude-3.7-Sonnet | Gemini-2.0-Flash | GPT-4.1 | Mean ± SD |
> |-----------|-------------------|------------------|---------|-----------|
> | **1** | 0.804 | 0.857 | 0.822 | 0.828 ± 0.027 |
> | **2** | 0.814 | 0.824 | 0.810 | 0.816 ± 0.007 |
> | **3** | 0.779 | 0.809 | 0.753 | 0.780 ± 0.028 |
> | **Mean ± SD** | 0.799 ± 0.018 | 0.830 ± 0.025 | 0.795 ± 0.037 | **0.808 ± 0.033** |
>
> ---

---

### Author Response · Authors · 2025-11-24
**Response to all reviewers**

Thank you all for the reviews and time devoted to our paper.  We have substantially revised the paper to address core concerns with revisions in blue text.

Below, we summarize the two major points of concern and our corresponding improvements.

---

# **Major Point 1 — Validity and Bias in LM-as-a-Judge**

A central concern across reviews involved whether using language models as judges introduces bias, circularity, or self-confirming behavior, especially because we use the same model as a generator and validator for the self-consistency scores.

Across the revisions, we have made the following explicit:

### **1. Self-consistency is a *minimum* validity requirement, not a full behavioral assessment**

Our framework defines **three-way consistency** among:
- A provider’s **specification**
- The provider’s **candidate model**
- The provider’s **judge model**

This is not a claim that the judge model is a perfect arbiter of behavior. Instead, it is a **minimal internal consistency check** because it is standard practice for the providers to use their models as a judge and they control all artifacts in question.

Failure under this most charitable setting reveals a fundamental internal contradiction: either the specification is unclear, the model does not follow it, or the judge model is miscalibrated. We now emphasize this point in Sections 4.1–4.2.

### **2. Bias and circularity are addressed through *cross-judge evaluation***

To ensure that results do not depend on any single judge model, we perform a full **cross-provider, cross-judge** analysis. In these experiments:
- Every candidate model is evaluated by **all three judge models** (GPT-4.1, Claude-3.7-Sonnet, Gemini-2.0-Flash).
- We compute **judge-agnostic adherence scores** by averaging across judges.

We have:
- Added a new paragraph ("How does using multiple judges influence results?")
- Clarified this distinction in Sections **4.1–4.3**,
- Moved Figure 5 discussion earlier and made cross-judge outcomes highly visible,
- Explicitly contrasted **self-consistency** (provider-internal) with **cross-judge** (provider-agnostic) metrics.



---

# **Major Point 2 — Human Validation of the Generation and Evaluation Pipeline**

All multiple raised questions about whether LM-generated prompts and LM-based judgments were appropriately validated. We revise our paper as follows

### **1. We moved all human validation results into the *main paper***

The human studies originally in Appendix F now appear in **Section 4.3**, including Tables 1–4. They evaluate both:
- the **quality and relevance** of prompts generated by TestMaker
- the **agreement** between LM judges and human raters

This ensures human validation. The human study on the relevance of prompts generated by the TestMaker was done during the rebuttal.

### **2. Prompt relevance human study (new)**

During the rebuttal, we conducted a new human study on **prompt relevance**:
- 19 specification statements subsampled
- 57 prompts evaluated
- Three qualified Prolific AI Tasker annotators
- Raters judged whether each prompt is relevant to evaluating the target behavioral specification

Results:
- **100%** of prompts judged *relevant* by majority vote
- Even the most pessimistic rater judged **89%** as relevant
- Full binary relevance table shown in paper


### **3. Human–model agreement vs. inter-human agreement**

Our original human validation study (now Section 4.3) shows:
- **Inter-human agreement:** 72.2% ± 3.4%
- **Human–model agreement:** 71.1% ± 4.7%
- **LM judge F1:** 0.808 vs. individual humans; 0.817 vs. majority vote

Crucially:
> **LM judges agree with humans at roughly the same rate that humans agree with each other.**

This finding directly addresses concerns about LM-judge reliability: the LM judges behave comparably to fully independent human raters.

### **4. All 2,360 prompts were manually reviewed by authors**

We clarified in Section 3.2 that *every* prompt—not just a subset—underwent manual review after generation, ensuring correctness and adherence to specification statements.

We have updated our paper so the main version has the results of the human study, but we will add a separate comment with the results of the human studies for convenience.

---

### Author Response · Authors · 2025-11-24
**Human Study Results**

---

# **Agreement Tables**

### **Inter-annotator agreement**

| Annotator Pair | Agreement |
|----------------|-----------|
| 1 vs 2 | 73.3% |
| 1 vs 3 | 75.0% |
| 2 vs 3 | 68.3% |

**Mean:** 72.2% ± 3.4%

---

### **Human–Model Agreement**

| Annotator | Claude-3.7-Sonnet | Gemini-2.0-Flash | GPT-4.1 | Mean |
|-----------|-------------------|------------------|---------|------|
| 1 | 70.0% | 78.3% | 73.3% | 73.9% |
| 2 | 73.3% | 75.0% | 73.3% | 73.9% |
| 3 | 65.0% | 70.0% | 61.7% | 65.6% |

**Overall Mean:** 71.1% ± 4.7%

---

### **F1 Score vs. Human Raters**

| Model Judge | Individual Mean | Majority Vote |
|-------------|-----------------|---------------|
| Claude-3.7-Sonnet | 0.799 | 0.787 |
| Gemini-2.0-Flash | 0.830 | 0.860 |
| GPT-4.1 | 0.795 | 0.804 |
| **Mean ± SD** | **0.808 ± 0.019** | **0.817 ± 0.038** |

---

### **Full F1 Score Matrix**

| Annotator | Claude-3.7-Sonnet | Gemini-2.0-Flash | GPT-4.1 | Mean ± SD |
|-----------|-------------------|------------------|---------|-----------|
| **1** | 0.804 | 0.857 | 0.822 | 0.828 ± 0.027 |
| **2** | 0.814 | 0.824 | 0.810 | 0.816 ± 0.007 |
| **3** | 0.779 | 0.809 | 0.753 | 0.780 ± 0.028 |
| **Mean ± SD** | 0.799 ± 0.018 | 0.830 ± 0.025 | 0.795 ± 0.037 | **0.808 ± 0.033** |

---

### *Results on Prompt Relevance Study*

| Annotator   | Relevant | Not relevant |
|------------|----------|--------------|
| Annotator 1 | 56       | 1            |
| Annotator 2 | 51       | 6            |
| Annotator 3 | 57       | 0            |

---

---

### Meta-Review · Area_Chair_zqBj · 2026-01-08

**Summary:**

The reviewers primarily questioned the validity of the proposed "three-way consistency" metric (circularity when using one model as both the candidate and judge), the lack of human anchoring to validate the automated "TestMaker" prompts and the "Judge" evaluations, and the absence of smaller-scale model baselines. Following the rebuttal, the authors conducted a series of experiments, including a human study that showed 72% agreement between the proposed metric and human validation. However, conceptual concerns regarding the interpretability and statistical significance of the three-way consistency metric remain.

Therefore, AC recommends rejection at this stage.

**Reviewer Concerns:**

Addressed:
1. Human validation: The authors introduced a new Section 4.3 with a human study involving 60 items. The results showed that the LM-Judge aligned with human annotators (~71.1%) at a rate comparable to inter-human agreement. Statistically speaking, a sample size of 60 is somewhat small, and the confidence interval reaches ±4.7%, which still presents certain limitations.
2. Evaluation with smaller model baseline: The authors added experiments for smaller models (Qwen2-7B-Instruct, Llama3.1-8B-Instruct).
3. TestMaker Bias: The concern that using GPT-4.1 as the sole TestMaker introduces bias was mitigated by the cross-judge results, which showed that Anthropic models still outperformed OpenAI models.

Still Outstanding:
1. Conceptual Validity of "Three-Way Consistency": While the authors clarified that this metric is a "minimal internal consistency check," the fundamental critique remains: the metric fails to isolate why a model fails (conflates "poor behavior" with "poor judging capability"). This makes the metric difficult to interpret as a pure measure of "adherence".
2. Statistical Significance: Reviewer YngW noted that the confidence intervals for the top models (e.g., Claude-3.7 vs. GPT-4.1) overlap significantly. The authors acknowledged this but argued it indicates "broad similarity." However, given that the paper frames itself as an auditing framework to identify gaps, the inability to statistically distinguish between major providers (due to the stochasticity of the judge/candidate) weakens the practical utility of the framework for ranking or comparative auditing.

**Reviewer Scores:**

- Reviewer brv4: remains 2.
- Reviewer g9U6: remains 4.
- Reviewer 7RhD: remains 4.
- Reviewer YngW: remains 6.

---

### Decision · Program_Chairs · 2026-01-26

Reject